# Galectins—Potential Therapeutic Targets for Neurodegenerative Disorders

**DOI:** 10.3390/ijms231911012

**Published:** 2022-09-20

**Authors:** Sapana Chaudhary, Sameer Chaudhary, Sakshi Rawat, Archana Kulkarni, Anwar L. Bilgrami, Asma Perveen, Badrah S. Alghamdi, Torki Al Zughaibi, Ghulam Md Ashraf, Mohammad Zubair Alam, Tabish Hussain

**Affiliations:** 1RASA Life Science Informatics, Pune 411052, India; 2Department of Entomology, Rutgers University, New Brunswick, NJ 08816, USA; 3Deanship of Scientific Research, King Abdulaziz University, Jeddah 21589, Saudi Arabia; 4Glocal School of Life Sciences, Glocal University, Saharanpur 247001, India; 5Department of Biochemistry, Faculty of Life Sciences, Aligarh Muslim University, Aligarh 202002, India; 6Department of Physiology, Neuroscience Unit, Faculty of Medicine, King Abdulaziz University, Jeddah 22252, Saudi Arabia; 7Pre-Clinical Research Unit, King Fahd Medical Research Center, King Abdulaziz University, Jeddah 21589, Saudi Arabia; 8King Fahd Medical Research Center, King Abdulaziz University, Jeddah 22252, Saudi Arabia; 9Department of Medical Laboratory Technology, Faculty of Applied Medical Sciences, King Abdulaziz University, Jeddah 22252, Saudi Arabia; 10University of Texas MD Anderson Cancer Center, Houston, TX 77054, USA

**Keywords:** galectins, neurodegeneration, neuroinflammation, Gal-1, Gal-3, Gal-9, targeted therapy

## Abstract

Advancements in medicine have increased the longevity of humans, resulting in a higher incidence of chronic diseases. Due to the rise in the elderly population, age-dependent neurodegenerative disorders are becoming increasingly prevalent. The available treatment options only provide symptomatic relief and do not cure the underlying cause of the disease. Therefore, it has become imperative to discover new markers and therapies to modulate the course of disease progression and develop better treatment options for the affected individuals. Growing evidence indicates that neuroinflammation is a common factor and one of the main inducers of neuronal damage and degeneration. Galectins (Gals) are a class of β-galactoside-binding proteins (lectins) ubiquitously expressed in almost all vital organs. Gals modulate various cellular responses and regulate significant biological functions, including immune response, proliferation, differentiation, migration, and cell growth, through their interaction with glycoproteins and glycolipids. In recent years, extensive research has been conducted on the Gal superfamily, with Gal-1, Gal-3, and Gal-9 in prime focus. Their roles have been described in modulating neuroinflammation and neurodegenerative processes. In this review, we discuss the role of Gals in the causation and progression of neurodegenerative disorders. We describe the role of Gals in microglia and astrocyte modulation, along with their pro- and anti-inflammatory functions. In addition, we discuss the potential use of Gals as a novel therapeutic target for neuroinflammation and restoring tissue damage in neurodegenerative diseases.

## 1. Introduction

The life span of humans is increasing globally, resulting in a dramatic rise in the incidence of neurological disorders (NDs). Neurodegeneration is the pathological feature of NDs in which neuronal cells lose their structure and function, causing disease development. Several factors lead to neurodegeneration; however, inflammation is proposed as a critical feature and one of the main inducers of degeneration in almost all NDs [1,2]. A perpetual response to neuroinflammation contributes to neurodegeneration, leading to progressive disease. 

Gals are carbohydrate-binding multifunctional proteins that are transcribed and translated by *LGALS* genes [3]. These are a family of soluble lectins with a small molecular weight ranging between 14 and 39 kDa. They have a consensus core sequence located in the carbohydrate recognition domain (CRD) and have an affinity for β-galactoside structures. The galectin CRD consists of approximately 130 amino acids, which creates a concave groove that defines the galectin-carbohydrate recognition site. The Galectin family has 16 members from Gal-1 to Gal-16, which can be divided into three subfamilies, based on the organization of the CRD [3]. Gals can bind branched glycans on glycoconjugates, resulting in lattice formation and lipid raft clustering [4,5,6,7,8]. These proteins can tie to sugar particles and can be present on different proteins inside or on the cell surface [9]. Gals are synthesized in the cytoplasm, from where they are directed into different cellular compartments and regions [10]. They are involved in regulating several cellular functions and have been shown to play a role in embryonic development, inflammation, immune response, metabolic diseases, RNA splicing, cell cycle regulation, motility, survival, cancer, and neurological disorders [2,3,4,14]. Recent observations suggest that Gals play an essential role in chronic neurodegeneration [11,12] and have a significant function in the adaptive immune response after acute central nervous system (CNS) injuries [13]. Because of its complex multifunctional properties, Gals hold tremendous therapeutic potential for various neurodegenerative diseases and other diseases, such as cardiovascular disorders [14], chronic inflammation, and cancer [12,15].

This review focuses on summarizing the immune-regulatory activities of three predominant members of the Gal family—Gal-1, Gal-3, and Gal-9—which are known to play a significant role in neurodegenerative ailments [16]. The inclusion criteria for the literature search were to incorporate studies describing the association of Gals with neurovegetative disorders and their role in neuroinflammation. We also included studies and reviews describing their potential as therapeutic targets. Neurodegenerative disorders such as Alzheimer’s disease (AD), Parkinson’s disease (PD), Huntington’s disease (HD), amyotrophic lateral sclerosis (ALS), and frontotemporal dementia (FTD) display similar characteristics, including native protein accumulation, neuronal degeneration, and cognitive and behavioral impairment [17]. All these NDs are identified by selective neuronal failure and are associated with primary proteinopathy, which tends to be much more complicated [18]. Gal-1 mRNA and protein levels have been shown to be higher in the spinal cords of SOD1 mice displaying phenotypes similar to ALS in humans. In addition, higher mRNA and protein expression of Gal-3 and Gal-9 has been observed in the spinal cords of SOD1 mice and sporadic ALS patients [19]. Gal-3 specifically has been identified as a biomarker in serum, plasma, and/or cerebrospinal fluid (CSF) in Alzheimer’s disease (AD), Parkinson’s disease (PD), and amyotrophic lateral sclerosis (ALS). It has also shown detrimental regulation of inflammatory responses in AD [11,20,21]. Further, moderate cognitive impairment in AD has been associated with Gal-9 [17,22]. Various physiological and neurotic diseases have been identified and verified using genetically altered or modified mice that are devoid of a specific Gal [19]. This information suggests that targeting Gals has a promising therapeutic potential to treat inflammatory and neurodegenerative disorders [3].

## 2. Physiological Role of Gals

Gals are an assembly of soluble β-galactoside binding proteins [23,24]. Based on the structure, Gals are broadly classified into three groups: prototypical (Gal-1, -2, -5, -7, -10, -11, -13, -14, -15, and 16), chimeric (Gal-3), and tandem-repeat (Gal-4, -6, -8, -9, -12) Gals [25]. They belong to a huge family of enzymes with changing sub-treaties within its family of proteins, which are identified by their binding energy for β-galactoside sugar [26,27,28,29]. They are also distinguished because they do not require a divalent positive ion for binding, share their primary structure motif, and have a unique structural fold [30], as shown in Figure 1 [9].

Different architectural forms contribute to various signaling functions (Figure 1) [9]. The prototypical Gal comprises only one CRD. In contrast, the tandem-repeat-type Gals consist of two identical CRDs bound together by a small polypeptide chain, 70 amino acids long, called a linker [16]. The chimera type, i.e., Gal-3, has an N-terminal non-lectin region of up to 130 amino acids in length, attached to a CRD [9]. The ability of any given Gal to recognize non-reducing terminal or internal galactosyl residues, because of its architectural form, is its most essential feature [9]. There are different exemptions, such as GRP (Gal-related protein), demonstrating no lectin qualities, the mammalian GRIFIN (galectin-related inter-fiber protein), having zero sugar restricting action, and Gal-10 (CLC; Charcot–Leyden crystal) perceiving mannosyl residues with higher specificity than galactosyl residues [9,27]. Regardless of a CRD with high closeness, each Gal exhibits extensive contrast in their polysaccharide, which specificity emerges from the unique structure and dynamics of the ligand-binding groove (LBG) [9]. Over the past several years, Gals have been investigated for various biological functions of immune and inflammatory responses and tumor improvement [3]. Different biological processes mediated by individual members of the Gal superfamily are summarized in Table 1 [9].

### 2.1. Gal-1

Gal-1 has been linked to neurodegeneration in many studies, and its expression is altered in various NDs [31,32]. Patients with neurological conditions exhibit higher levels of Gal-1 compared to healthy control subjects [33]. Gal-1 is a pro-inflammatory cytokine protein that regulates tumor suppressor growth, adhesion, signaling, cell differentiation and development, the immune system, and host micro-pathogen interactions [34,35]. Gal-1 also plays a vital role in the embryonic development of sensory neurons and their synaptic connections in the neural cord, as seen in the growth of the neural system of mice’s olfactory bulb [25,36]. Gal-1 has been linked to a variety of NDs and has been linked to increased cell expression of the regenerative progress injury site, in addition to playing a protective function during reperfusion injury [25,37].

Gal-1, a glycan-restricting lectin encoded by LGALS1 in humans, has long been thought to be an essential regulator of microglia activation [38]. Gal-1 encourages M1 microglia deactivation and upregulates the phenotype of M2 microglia activation, which delays neurodegeneration and increases neuroprotection in the long run [38]. Gal-1 is generated by astrocytes, which primarily serve as a biochemical aid to endothelial cells in forming the b3 (blood–brain barrier) [38]. In reality, during the acute and chronic phases of experimental autoimmune encephalomyelitis (EAE), astrocytes express high levels of Gal-1. EAE is an infectious model seen in animals that can be caused by hypersensitivity reactions with antigens such as myelin proteins or myelin protein peptides. Gal-1-deficient mice showed abundant M1 microglia and reduced immunoreactivity against nerve and axon markers, demonstrating that Gal-1 switches off M1 microglia reaction and prevents neuronal injury in EAE [38]. Because Gals recognize galactose and oligos containing galactose, most drug development efforts have concentrated on the synthetic modifications of polysaccharides [25]. Each polysaccharide scaffold has a significant advantage over another when manipulating the aspects of affinity, stability, or selectivity [25].

### 2.2. Gal-3

Gal-3 is the most well-known member of the Gal family, with an extended N-terminal region made up of tandem repeats of short amino acid segments (a total of around 120 amino acids) and a C-terminal CRD [16]. Gal-3 is connected to several intracellular functions by collaborating with intracellular proteins that serve as mediators. Its phosphorylation ability at the Ser6 (Serine-6) and Ser12 residues is important [34]. Gal-3, a carbohydrate-binding chimeric-type lectin encoded by the LGALS3 gene in humans, is gaining notoriety as a “rogue enzyme molecule” [19]. Furthermore, forensic tests of ALS patients revealed that Gal-3 levels in spinal cords and brain stalks were significantly higher than normal controls [39]. Gal-3 has been found to participate in several cellular functions, such as apoptosis, cellular adhesion, cell fragmentation, cell differentiation, immunity response, and inflammation [19,25,40,41]. Despite studies that show that Gal-3 is involved in the neurophysiological ALS process, it has not yet been clarified whether Gal-3 provides a protective or antiviral reaction to ALS [19]. Gal-3 is a secretory protein that has been shown to be expanded when compared to controls [39]. A study has shown that ALS patients have approximately two times the Gal-3 in CSF compared to NCs, similar to other neurological conditions, including stroke and dementia [39]. Plasma levels of Gal-3 further enhanced the development of ALS, which showed it could have a neuroprotective effect [39]. In future clinical trials, observations of disease processes are needed to approve these competitive biomarker proteins as evident biomarkers of ALS [39]. In addition, LGALS3 genetic variation encoding Gal-3 have been associated with systemic sclerosis patients and cognitive functions at old age [42,43]. The role of Gal-3 in different neurodegenerative disorders is shown in Figure 2.

## 3. Galectins as Extracellular Binding Proteins

Gals are synthesized as cellular molecules without signaling proteins and are metabolized by an unconventional mechanistic pathway that probably involves exomes [26,44,45]. Metabolized Gals interact with the collagen fibers and a wide range of cellular glycoproteins, mitigating protein synthesis and cellular and intracellular signals stimulation, or can enter the cell via endocytosis, where they play roles in glycoprotein post-Golgi trafficking [46]. The extracellular trafficking and signaling processes of cellular glycoproteins bind Gals to regulate cellular proliferation, cell differentiation, survival, fragmentation, adhesion, migration, and proteostasis, depending on the Gal and the cellular background [47,48,49,50]. In contrast to cell regulation, Gals connect with many cell surface binding glycoproteins by the specific interaction of ligands with signaling molecules. Thus, they build more global cellular regulation and sensing and provide a different context for the ligand binding system [50,51].

## 4. Galectins Auto-Antibodies

As a possible treatment for secondary-progressive multiple sclerosis (MS) [52], anti-Gal-8, the autoantibodies against Gal-3, have been proposed as an early prognostic marker for relapsed MS [53]. The blood–brain barrier’s direct interaction with Gal-3 is opened up in microvascular endothelial cells, while the immunomodulatory and neuroprotective function in Gal-8 is neutralized by antimicrobial anti-Gal-3 antibody [52,53]. Initially, anti-Gal-8 antibodies were found in lymphopenia, autoimmune diseases, and sepsis in systemic lupus erythematosus, where their microorganism contribution is unsolved [54]. These function-blocking proteins inhibit Gal-8 interactions with β1 and LFA1 integrins [55,56], as well as T-cell adhesion and apoptosis in activated T cells, including TH17 cells [53], and hippocampal neuron viability in primary culture [52]. Gal autoantibodies can be seen as potential therapeutic agents and factors that interfere in Gal treatment [55].

## 5. Gals and Neurodevelopment

### 5.1. Gal-1

Gal-1 mediates the deactivation of M1 microglia and activation of a phenotype of M2 microglia, which prevents neuronal degenerative activity and promotes neuronal protection [38]. Gal-1 is synthesized by astrocytes, primarily acting as a neuronal defensive support system for the b3 endothelial cells. During both the acute and chronic phases of EAE, astrocytes expressed elevated levels of Gal-1. By displaying abundant M1 microglia and reduced immunoreactivity against neuronal and axonal markers in Gal-1-deficient Lgals1/mice, Starossom et al. demonstrated that Gal-1 deactivates M1 microglia and prevents axonal damage in EAE. Multivalent lectin interactions with glycan possibly trap glycoprotein receptors at the cell surface and prevent their endocytosis, prolonging intracellular signaling due to increased responsiveness to extracellular signal inputs [57,58]. Gal-1 linked to the core 2-branched O-glycan of CD45, a phosphatase that transduces inhibitory signals for M1 activation [59], promoted glycoprotein retention on the surface of microglia cells and prolonged and augmented its phosphatase activity. Starossom et al. also showed that exogenous Gal-1 significantly reduced EAE disease severity and microglia activation in the spinal cord in vivo, indicating that targeting the Gal-1-glycan axis may be a new therapeutic strategy for diseases involving oxidative stress neurodegeneration, such as MS. Because of the pathological similarities between the rodent EAE model and MS, the hypothesis of MS pathogenesis based on activated T cell destruction of myelin was widely accepted [60]. However, human MS differs from mouse EAE in terms of the cause or critical aspects of the initiating factors; the role of microglia in human MS is thought to be more significant than in EAE [61]. When T cell density in cortical MS lesions was measured, there were few inflammatory cells, no perivascular cuffs, and no phagocytic macrophages, but there were many activated microglia [62]. As a result, these findings add to the growing body of evidence that microglia, rather than T cells, play a direct role in cortical demyelination in human MS. Thus, the neuroprotective function of Gal-1 in the CNS of human MS should be validated (e.g., a negative association between the amount of Gal-1 and disease progression). The molecular mechanism by which Gal-1 expression in astrocytes is physiologically regulated can lead to a better understanding of MS pathogenesis. Although the research by Starossom et al. offered new insights into the cause of MS, clinical use of exogenous Gal-1 as a therapeutic tool for MS can still meet significant challenges.

Gal-1 has been linked to T cell expression and has the widest distribution in mammalian tissues, where it was relatively abundant [63]. Gal-1 CRD’s β sheet motif has a high degree of structural homology with other Gals, such as Gal-3. While no evidence of functional redundancy among Gal-1 and Gal-3 has been found in Wistar rat experiments, this is thought to be due to variations in tissue distribution and subcellular localization rather than differences in carbohydrate-binding specificities [64]. Nonetheless, a number of Gal family members have been related to the regulation of CNS macrophages and microglia in various ways. In a model of traumatic brain injury, Gal-3 is upregulated in microglia, favoring myelin phagocytosis. Gal-9, on the other hand, stimulates innate immunity by signaling via Tim-3 on CD11b+ CNS cells [64].

Consequently, oligodendrocytes, which respond to myelination, express high levels of Gal-3 and Gal-4. Interestingly, these studies indicate that a small but significant difference in carbohydrate specificities among Gals may result in opposite effects on MS pathogenesis. In the field of MS study, comprehensive and detailed profiling of Gal-carbohydrate ligands will be needed. Furthermore, Gal-1’s association with ligands other than CD45 on microglia should be carefully considered. When developing a therapeutic strategy for Gal-1, its functional specificity in interacting with CD45 will be particularly significant. Alternative antibodies targeting CD45’s central two-branched O-glycan will enhance Gal-1’s nonspecific identification. Regardless, the current research by Starossom et al. (2012) emphasizes the function of Gal-1 in adjusting immune balance from neurodegenerative diseases to neuroprotection, which may aid in our understanding of human MS pathogenesis.

### 5.2. Gal-3 

Gal-3 is expressed in glial cells and neuronal tissues in various brain regions in normal adult rats [65]. Gal-3 co-expressed with antigens for various cell types in the rat brain, including neuronal nuclear antigen (NeuN), glial fibrillary acidic protein (GFAP), and ionized calcium-binding adapter molecule 1 (IBA1). The telencephalon (some segments of the cerebral cortex with differences in the laminar distribution and regions of the amygdala, basal ganglia, and septum), diencephalon (thalamus and hypothalamus), brain stem, and cerebellum (mesencephalon, rhombencephalon, myelencephalon, and cerebellum) all showed immunoreactivity for Gal-3 [65]. According to Comte et al., Gal-3 plays a role in normal mouse brain neurodevelopment [66]. Gal-3 influences neuroblast migration from the subventricular zone (SVZ) to the olfactory bulb (OB) through the rostral migratory stream (RMS) [66]. The migration of neuroblasts was disrupted in Gal-3 knock-out (KO) mice due to a decrease in migration speed and straightness. In the absence of Gal-3, increased phosphorylation and activation of the epidermal growth factor receptor (EGFR) is one of the potential mechanisms. Keratinocytes with Gal-3 deficiency have slower migration and lower EGFR expression on the surface [58,67]. Gal-3 plays a vital role in oligodendrocyte cell differentiation as well as the integrity and function of sheath muscles [68]. According to an in vitro analysis, Gal-3 is expressed by oligodendrocytes at various stages of differentiation. According to in vivo studies, Gal-3 mediates oligodendrocyte differentiation by microglial cells and astrocytes [68]. Furthermore, when comparing Gal-3 KO mice to WT mice, electron microscopic examination of myelin revealed that the myelination mechanism was disrupted in Gal-3 KO mice. Thomas and Pasquini showed that Gal-3 mediated glial crosstalk and drives oligodendrocytes differentiation [69]. According to Thomas and Pasquini, Gal-3-mediated glial crosstalk drives oligodendrocyte differentiation [69]. One of the most common causes of neurological disorders in infants is asphyxia, which is accompanied by hypoxic and ischemic brain damage [70,71]. Hypoxia and ischemia cause the immune system to become activated, resulting in an increase in the development of pro-inflammatory immune cells and antioxidants [72,73,74]. In this neuroinflammation, microglial cells play a critical role. They also contain Gal-3, a pro-inflammatory protein produced in hypoxia and ischemia [75,76]. Doverhag et al. observed increased expression of Gal-3 RNA 8 h, 24 h, and 72 h after injury in newborn mice subjected to neonatal hypoxia/ischemia, and Gal-3 colocalized with Iba-1 in activated microglia close to the injury. Furthermore, Gal-3-deficient mice were covered relative to their WT littermates, as neuronal cell volume loss and regional damage of the hippocampus and striatum were significantly reduced. Although there was no statistically significant difference in microglia accumulation between Gal-3-deficient mice and WTs, total matrix metalloproteinase 9 (MMP-9) protein levels in WTs were significantly higher, indicating that Gal-3 may modulate microglia phenotype and wound attenuation in Gal-3-deficient rodents [77]. According to a study on C57Bl/6 Ros production in KO mice, hypoxic brain injury increased Gal-3 levels in NADPH KO mice relative to WT mice and in the injured hemisphere compared to the uninjured hemisphere in both KO and WT mice [78]. According to Pesheva et al., Gal-3 expression in neurons depends on the presence of nerve growth factor (NGF). The authors used neonatal dorsal root ganglion (DRG) neurons to investigate the effects of NGF, BDNF, and neurotrophin-3 on Gal-3 expression levels and cell types that express Gal-3. They discovered that NGF-activated DRG neurons and macrophage-like cells increased Gal-3 expression, implying a role in neurite outgrowth and neural cell adhesion [79,80]. The activation of the tropomyosin receptor kinase A (TrkA) receptors was also proposed as a molecular mechanism. At the same time, a later study found that modulation of Gal-3 expression was mediated through Ras/MAPK-related signaling pathways [79,81]. The same group of authors, on the other hand, found that staurosporine (a protein kinase inhibitor) induced Gal-3 secretion in PC12 cells after 1–5 days, which was unaffected by Ras/MAPK pathway inhibitors, indicating a Ras/MAPK-independent mechanism for Gal-3 expression control [81]. Umekawa et al. contrasted inflammatory responses to hypoxia/ischemia-induced brain injury in the immature and adult hippocampus based on discrepancies among resident microglia and macrophages from circulation [74]. Based on the previous findings, the authors concluded that resident microglia activated earlier in the immature brain and triggered a more pronounced inflammatory response than infiltrating blood-derived macrophages [74]. Furthermore, Gal-3 expression was higher in immature brains, which may be attributed to a more severe inflammatory response in newborn organisms [74]. In contrast, it was found that up-regulation of Gal-3 in neonatal mice and rats affected by transient occlusion of the middle cerebral artery resulted in a focal stroke. In Gal-3-deficient mice, tissue loss was more significant than in wild-type mice [76]. Furthermore, certain cytokines and chemokines were altered in Gal-3-deficient mice 72 h after brain damage was induced, with lower levels of interleukin 6 (IL-6) and granulocyte-colony stimulating factor (G-CSF) and higher levels of macrophage inflammatory protein 1 (MIP-1a) and MIP-1b. Gal-3 has a novel inflammation-independent role in regulating astrogenesis by altering bone morphogenetic protein (BMP) signaling [82]. Because periventricular regions of the brain are particularly vulnerable to hypoxic ischemia, the authors concentrated on the postnatal lateral subventricular zone. They used electroporation to increase or decrease Gal-3 expression in vivo and nucleofection in vitro [82]. During postnatal forebrain production, the subventricular zone is the primary source of glial cells, and Gal-3 deficiency decreases gliogenesis, while Gal-3 increase has the opposite effect. In addition, Gal-3 binds an alpha (BMPR1α), activates BMP signaling, and thus regulation basal gliogenesis [82].

### 5.3. Gal-4

The axon is essential as a neuron output channel for the conduction of the nerve and rapid transmission of nerve impulses. For axon growth neurons, Gal-4 is necessary [83]. Axon growth has been proven by increasing neural cell adhesion (NCAM) L1’s cluster number and size presence in the axon membrane [83]. NCAM L1, a postmitotic neuronal axon glycoprotein, regulates outpatient neuritis, nervous conduction, and branches by homophilic L1-L1 interactions [84,85]. Gal-4 facilitates the organization of L1 membrane clusters by binding to N-acetyllactosamine (LacNAc) at the branch ends of L1 N-glycans, which is a regulator of synaptic glycoprotein axonal transport [83]. Gal-4 is essential to properly organize and function L1 in the central nervous system (CNS). In the expression of myelination of axons, Gal-4 is also important, and its expression at the start of myelination is downregulated [78]. Myelin is a lipid-rich membrane formed by oligodendrocytes (OLGs) that wraps nerve axons, forming a multilamellar insulating sheath with plasticity and high cognitive functions in the CNS [79]. We divided the myelination process into three parts based on the position of Gal-4 in the CNS: Nonmyelinated neurons could produce and release Gal-4, which would then bind to cell surface receptors expressed by pre-myelinating OLGs [80]. Gal-4 can partially promote the dedifferentiation and proliferation of OLGs by binding to specific receptors [80]. Gal-4 is released in OLGs, which promotes the expression of the myelin basic protein (MBP) gene, likely by moving from cytoplasmic to nuclear localization during OLG maturation. MBP is a major protein of the myelin sheath that keeps the structure and function of CNS myelin stable. It is found on the myelin serosal surface. Gal-4 is thought to regulate MBP expression by binding to the glycosylated moiety of transcription factor Sp1 alternately and then stimulating Sp1 stability and possibly nuclear cell localization [12]. Sp1 can stimulate the MBP promoters in the cell of OLGs [3] [82,83]. Gal-4 expression was downregulated at the initiation of myelination, providing a favorable environment for OLG differentiation and maturation. The mature OLGs were able to spirally surround axons with their cell membrane, aiding in the development of the myelin sheath. Sulfatide, a high binding affinity that interacts with ligands for Gal-4, has recently been developed as an inhibitor of sulfatide axon outgrowth, leading to the idea that Gal-4 regulates myelination through sulfatide binding interaction [86,87]. Gal-4 serves as a novel key regulator of OLG differentiation in neurons and facilitates axonal myelination in the central nervous system [85]. Figure 3 shows the of role of Gal-1, Gal-3, and Gal-4 in neurodevelopment. 

## 6. Gals in Neuroinflammation

Gals influence several immune system responses in astrocytes and microglia, such as cellular death, cell reactivation, cell adhesion, and the release of cytokines [79]. The Gal family has therefore been suggested to perform pro- or anti-inflammatory activities in initiating, performing, and resolving acute or chronic inflammatory responses. The molecular impact of Gals on the regulation of inflammatory response and their capacity to remodel damaged neuronal cells has been examined in studies on neurodegenerative diseases [80,81]. Neuroinflammation is orchestrated by CNS cells (neurons, astrocytes, and microglia), pattern recognition receptors (PRRs), cytokines, chemokines, complement proteins, peripheral immune cells, and signaling pathways [88]. The pro-inflammatory cytokines TNF- and IL-1, as well as the toll-like receptor (TLR) pathways, which cause NF-kB activation, are involved in neuroinflammation [89]. The NF-kB pathway plays a key role in inflammation and adaptive immune response activation through the expression of pro-inflammatory cytokines, chemokines, and gene expression of cell molecules [74]. Gal-1 expression is regulated by the NF-kB transcription factor, which has an anti-inflammatory effect [76]. Gal-1 induces apoptosis in Th1 immune cells at the peripheral stage, initiating IL-10 expression and the negative suppression of pro-inflammatory cytokines, resulting in a restricted immune system [82]. Gal-1 activity induces astrocyte differentiation and modulates their proliferation during the first step of the immunological response in the CNS. Gal-1 induces the expression of a brain-derived neurotrophic factor in differentiated astrocytes, causing astrocytes to undergo a reactive response and become neuroprotectors under most conditions, whereas severe astrogliosis is harmful and corresponds to neuronal tissue death [83].

### 6.1. Gal-1

Gal-1 is a pivotal regulator of M1 microglial activation that targets the activation of p38MAPK, CREB, and NF-kB-dependent signaling pathways and hierarchically suppresses downstream pro-inflammatory mediators such as iNOS, TNF, and CCL2 [38]. Gal-1 enhances the M2 microglial phenotype by modulating p38 mitogen-activated protein kinase (p38MAPK), cAMP response element binding (CREB), and the NF-kB signaling pathway, which promotes neuroprotection by inhibiting microglia [38]. The reduced form of Gal-1 promotes nerve cell progenitor cellular proliferation in the subventricular zone (SVZ), whereas its expression regulates neurogenesis in the SVZ [90]. Gal-1 binds to the core of CD45 2 O-glycans, allowing it to stay on the microglial cell surface longer, leading to an increase in its phosphatase activity and inhibitory role. Gal-1 is highly expressed during the acute phase of EAE, and knocking it out causes significant inflammation-induced neurodegeneration [38]. The adoptive transfer of Gal-1-secreting astrocytes or administration of recombinant Gal-1 suppresses EAE through microglial deactivation mechanisms [38]. Gal-1 glycan interactions are essential regulators of microglial activation, brain inflammation, and neurodegeneration, with important therapeutic implications for multiple sclerosis [38]. Gal-1 therapy causes microglia polarization to move from M1 to M2, which can help to avoid neurodegeneration and encourage neuroprotection [12]. These findings suggest that Gal-1 treatment reduces neuroinflammation in the CNS microenvironment by modulating the NO-arginase network in microglia and, as a result, may play a neuroprotective role in HAND [12].

Furthermore, the therapeutic ability of Gal-1 may be enhanced by conjugating it to gold nanoparticles (Au-NP), resulting in a multivalent nanogold Gal-1 (Au-Gal-1) complex with improved therapeutic translational efficacy compared to free Gal-1 due to increased payload influx [12]. In the early phase I clinical trials, GR-MD-02, a Gal-3 inhibitor, was well tolerated in patients with fatty liver disease/non-alcoholic steatohepatitis (NASH) and advanced liver fibrosis [91]. This suggests regulating Gal-1 expression is a novel therapeutic strategy for various neurodegenerative diseases [86].

### 6.2. Gal-3

According to recent research, Gal-3 appears to play a role in neuroinflammation and neurodegeneration [92]. Gal-3 knockout mice developed in the C57Bl/6 model do not display a distinct phenotype compared to WTs [86]. EAE is a widely accepted model for various sclerosis that describes pathological changes in humans [93]. In C57Bl/6 mice, deleting the Gal-3 gene reduces EAE [94]. This was due to antigen-presenting cell activation and resulting inflammatory response attenuation in the CNS. The incidence of diseases was substantially reduced in Gal-3-deficient mice after immunizing WT and Gal-3 mice with myelin oligodendrocyte glycoprotein peptide (MOG).

Furthermore, the development of pro-inflammatory cytokines such as IL-17 and IFN was reduced in these mice, while dendritic cells (DC) expressed IL-10 and exhibited Th2 polarization. Gal-3 is also associated with IL-4 mediated macrophage cell alternative polarization, so its role in EAE could be understood by its role in microglia activation and cell proliferation [16,94]. Microglia are immune-competent brain cells that play significant roles in different nervous-system homeostatic processes and in various pathological conditions that affect brain homeostasis [95]. Microglia activation can be a double-edged sword, with its profiling relying on a variety of factors. According to Reichert and Rotshenker, Gal-3 can play a role in the pathogenesis of EAE [96]. First, they discovered that mice with EAE had higher levels of Gal-3 expression in macrophages and microglia in the CNS. They were able to inhibit EAE by using Copolymer 1 as an immunomodulator.

Copolymer 1 stimulates antigen-specific Th2 response and increased secretion of IL-10, which lowers the development of pro-inflammatory cytokines and Gal-3; this was caused by the low activation of microglia and macrophages. It has also been suggested that Gal-3 is a key activator of modified myelin phagocytosis, which is needed for its regeneration in Wallerian degeneration. In contrast to microglia that do not phagocyte the myelin sheath, microglia that phagocyte myelin have higher levels of Gal-3 expression [97,98,99]. In Gal-3-deficient mice, Wallerian degeneration after sciatic nerve damage was associated with a significant increase in inflammatory cells IL-1 and TNF-, as well as up-regulation of toll-like receptors (TLR) 2 and 4 [100]. The C57Bl/6 Wistar model of focal cortical EAE formed large lesions with a high number of Gal-3-positive inflammatory cells after being immunized with MOG and receiving an intracerebral treatment of tumor necrosis factor (TNF) and interferon (IFN) [101]. These cells were divided into two major categories: Gal-3-projection positive cells, macrophages derived from microglia, and Gal-3-projection-free positive and monocyte-derived macrophages. In an animal model of EAE induced by pathogenic T-cell transfer, the latest information on cell and stage-specific expression of Gal-3 revealed increased expression of Gal-3 in microglia with increased phagocytic activity in the spinal grey matter as the disease progressed [102]. In addition, Gal-3 expression increased in spinal white and pia material microglia and macrophages during disease progression, while Gal-3 was positive for the subpopulation of Schwann’s nerve roots. Gal-3 expressing tissues were eliminated from parenchymas during rehabilitation and confined to the pia mater and ventral nerve roots. These findings show that Gal-3 can play a neuroprotective element in brain pathology, which means that Gal-3 may have pro- and counter-inflammatory impacts on the CNS. Their role seems to depend on the cell type and location of Gal-3 cells.

Furthermore, the effects of Gal-3, when released on immune cells and overexpressed in target cells, can be contrasted in type 1 and type 2 diabetes. Gal-3 deficiency causes type 1 diabetes to be less severe due to its absence in immune effector cells, as we previously demonstrated [102]. In type 2 diabetes, however, pancreatic cells are protected from inflammatory attack by intracellular genetic overexpression of Gal-3 (knock-in mice) [103]. Gal-3 is involved in the pathogenesis of CNS viral infections. Intracerebral immunization of the Junin virus triggered encephalitis in C57Bl/6 mice, and it was discovered that active microglia and astrocytes release Gal-3 [104]. Gal-3 expression increased in the cerebral cortex of C57Bl/6 and SJL/J mice after infection with Theiler’s murine encephalomyelitis virus (TMEV) [105]. Gal-3 deletion in C57Bl/6 mice also decreased the number of mobilized immune cells and the inflammatory response after TMEV infection, accompanied by partial restoration of SVZ proliferation and a rise in SVZ progenitor cells.

### 6.3. Gal-4

In the detergent-insoluble complexes known as fat rafts in intestinal epithelias, Gal-4 forms specific high diluted atoms with brush border enzymes [106]. RNA interference was used to extract Gal-4 from a human colon adenocarcinoma cell line to study its function in lipid rafts [106]. Apical membrane protein receptors were stuck intracellularly in Gal-4-depleted cells, indicating a problem with the sorting of apical membrane proteins [106]. Sulphatides containing “long-chain hydroxylated” fatty acids, which are abundant in lipid rafts, have been discovered to have a strong affinity for Gal-4 [107]. According to this finding, Gal-4 can play a significant role in apical protein supply by interacting with sulphatides to encourage lipid raft clusters [107]. In patients with inflammatory bowel diseases, Gal-4 was related to the disease of colitis. The gal-4 antibody is used to improve the condition of the moustache with mild intestinal inflammation [108].

### 6.4. Gal-9

Gal-9 was found to be upregulated in multiple sclerosis patients’ brains, suggesting it is upregulated through inflammation [109]. Human Gal-9 is a chemoattractant for eosinophils produced by T cells [110]. Surface Gal-9’s N- and C-terminal CRDs bind with ligands on the eosinophil surface that are similar or identical [111]. TIM-3, a Gal-9 receptor expressed on intrinsic immune cells, promotes tissue inflammation [111]. Gal-9, expressed by a variety of tumor cells, plays an important role in tumor resistance by controlling the survival, multiplication, and migration of both tumor and immunological cells in the tumor microenvironment [112]. A variety of diseases, their pathogenesis, and the potential therapeutic effects of various Gals are summarized in Table 2 [113].

## 7. Role of Gals in Neuronal Re- and Degeneration

### 7.1. Gal-1

Gal-1 targets activated microglia, increasing myelin phagocytic capacity and causing a shift toward an M2 phenotype, which leads to oligodendroglial differentiation [114]. Gal-1 induces neuronal degeneration because mice that lack Gal-1 expression display abnormal cellular differentiation in the brain [32]. Considering the scenario, using a Gal-1 inhibitor may be useful for treating neuronal degeneration. However, several studies have indicated that oxidized Gal-1 promotes neurite outgrowth [115]. Regardless of this ambiguity, if endogenous Gal-1 is shown to cause neuronal degeneration, Gal-1 inhibitors can potentially be used in either situation because it will be possible to develop inhibitors that do not inhibit oxidized types [115]. Traditional protein formulation methods have drawbacks in terms of pharmaceutical properties (high molecular weight, proteolytic degradation, rapid clearance, limited cellular uptake), structural fragility (physical/chemical inactivation during formulation and storage), and intrinsic immunogenicity [116]. However, appropriate nano carriers, such as nano formulations, can be used to transport active molecules to the desired site, targeted delivery, and controlled release approaches [117]. The therapeutic efficacy can be improved by releasing bioactive compounds only when they are stimulated (physical/chemical). In the case of Gal-1, the occurrence of different oxidative states and the presence of a monomer-dimer equilibrium, both of which documented differential activity in normal and pathological processes, present additional challenges in the development of therapeutics [118].

### 7.2. Gal-3

Gal-3 is produced by astrocytes, macrophages, microglia, endothelial cells, and Schwann cells in the CNS and peripheral nervous system (PNS) [119]. The majority of Gal-3 is co-restricted with bioactive markers for microglia and macrophages in variable segments from ALS spinal cords and brainstems [19]. As per the knowledge, based on a few known activities of this protein, a hypothesis regarding the possible organic component of Gal-3 in the pathogenesis of ALS can be established [39]. Plasma Gal-3 levels have been positively correlated with disease length, indicating that plasma Gal-3 may be an important element associated with ALS [39]. Plasma Gal-3 is an interesting factor linked to ALS that could be useful in assessing disease progression. Even though the role of Gals in ALS is unclear, given their secretory nature and detectable level in CSF, they can be used as a potential biomarker for ALS and possibly in future clinical trials for therapeutic use.

## 8. Conclusions

Galectins have emerged as potential therapeutic agents for many diseases due to their ubiquitous presence and involvement in various cellular functions. It is therefore important to understand the functions connected with the components of Gals. Despite several reports suggesting the potential of Gals as promising therapeutic targets, one of the limitations of targeting galectins is their significant involvement in several biological functions. In addition, the degree of reproducibility and specificity within the Gal family needs to be carefully addressed. To summarize, significant research has been carried out on members of the Gal superfamily with specific emphasis on their role in NDs. However, a great deal still needs to be unraveled. Given the emerging role of Gals in various intra-and intercellular pathways related to the pathogenesis and progression of NDs, as well as the success of the use of Gals as a target in the treatment of other human ailments, it will be worthwhile to recognize and exploit their potential as novel therapeutic targets against neurodegenerative disorders.

## Figures and Tables

**Figure 1 ijms-23-11012-f001:**
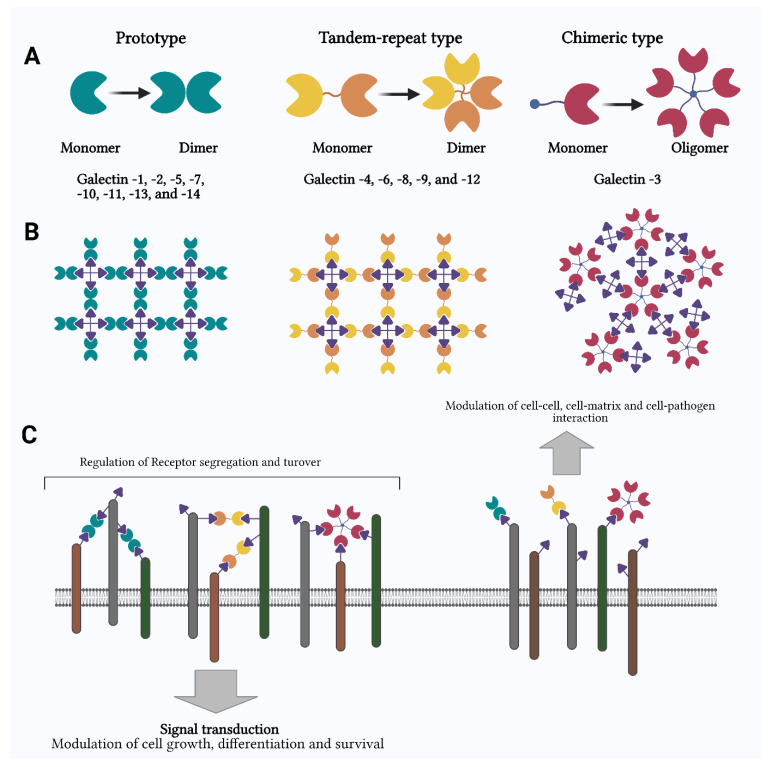
Architectural and functional schematic of galectins (**A**–**C**).

**Figure 2 ijms-23-11012-f002:**
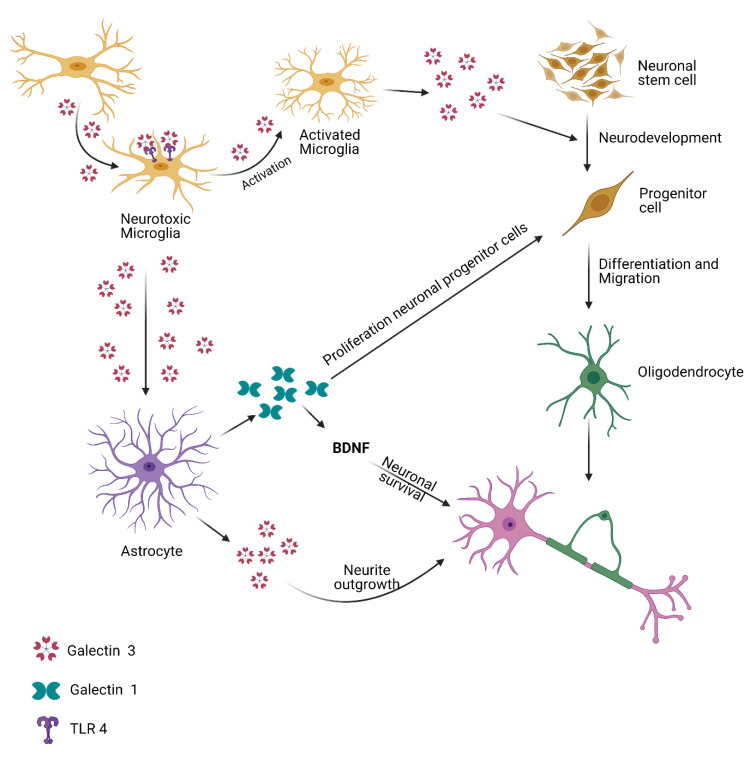
Role of galectins in microglia and astrocyte activation.

**Figure 3 ijms-23-11012-f003:**
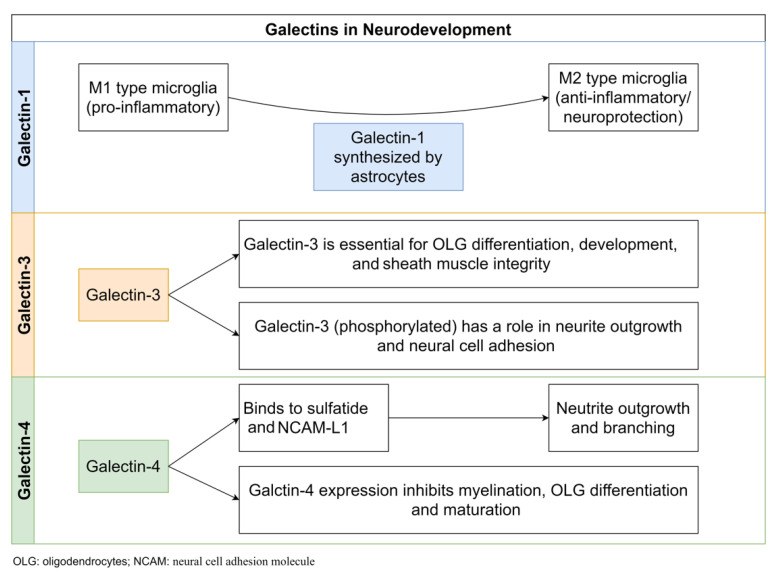
Role of Gals in neurodevelopment.

**Table 1 ijms-23-11012-t001:** Selected biological functions assigned to different members of the galectin family.

Galectin	Individual Functions
Galectin-1	▪ Incites apoptosis in initiated T-cells ▪ Smothers Th1 and Th17 insusceptible reactions ▪ Adds to the immuno-suppressive movement of administrative cells ▪ Adds to elective actuation of macrophages ▪ Favors tumor development and relocation ▪ Advances muscle cell separation ▪ Upgrades axonal recovery ▪ Advances expansion of neural undeveloped cells ▪ Adds to pre-m-RNA joining ▪ Controls platelet work
Galectin-2	▪ Incites apoptosis in initiated T-cells ▪ Ties to lymphotoxin
Galectin-3	▪ Contribute to pre-m-RNA splicing ▪ Induces apoptosis of T-cell (extracellular) and protects T-cell from apoptosis ▪ Potentiates eosinophil migration and promotes neutrophile transmigration and degranulation ▪ Favours Th2 responses (extracellular) and Th1 responses (intracellular) ▪ Promotes pro-inflammatory cytokine secretion. ▪ Favors mast cell degranulation
Galectin-4	▪ Takes an interest in the arrangement of lipid pontoons. ▪ Instigates the union of IL-6 by CD4+T cells. ▪ Profoundly communicated amid growth movement.
Galectin-7	▪ Partakes in keratinocytes separation. Khunmod ▪ Intercedes the master apoptotic impacts of p63 in keratinocytes. ▪ Over-communicated in metastatic mouse lymphoma cells.
Galectin-8	▪ Tweaks collaboration of integrins with extracellular lattice ▪ Upgrades glue properties of neutrophiles ▪ Smothers movement of growth cells ▪ Tweaks endocytosis of cell-surface receptors
Galectin-9	▪ Tweaks collaboration of integrins with extracellular lattice ▪ Upgrades adhesive properties of neutrophiles ▪ Smothers movement of growth cells ▪ Tweaks endocytosis of cell-surface receptors ▪ Chemoattractant of eosinophiles
Galectin-10	▪ Has an important role in T-regulatory cells function
Galectin-12	▪ Prompts apoptosis of adipocytes ▪ Partakes in the control of cell cycle movement ▪ Partakes in adipocyte separation

**Table 2 ijms-23-11012-t002:** Pathogenesis of numerous diseases and potential therapeutic effects of different galectins.

	Pathogenesis/ Pathophysiology	Galectin Expressed	Therapeutic Potential
Cancer	▪ Carcinogens weaken cell walls, microbes enter, divide out of control ▪ Cancer cell blocks apoptosis, lure/recruit macrophages ▪ Macrophages M1 → M2 triggers inflammation, angiogenesis ▪ Cancer cells form a tumor using angiogenesis ▪ Cancer cells metastasize	Gal-1 Gal-3	▪ Gal protein form a protective shield across the cancer cells (tumors), causing hindrance and preventing attacks by the immune T-cells and, also, kill T-cells in return. However, with the prologue of small non-digestible sugar molecules, the galectins can bind to those sugars and, as a result, break the shield and leave the cancer cells exposed and opened to be attacked by the immune T-cells.
Cardio-vascular diseases (CVD)	▪ Coronary vessel occlusion (atherosclerosis with thrombus) ▪ Dynamic changes in the plaque ▪ Plaque disruption → PLTS aggregation → thrombus and VC ▪ Necrosis (coagulation)	Gal-1 Gal-3 Gal-9	▪ Gal-1 is positively associated with angiogenesis, as is Gal-3 and -8 by different mechanisms. ▪ Gal-1, -3, -9 are all found to be pro-atherosclerotic. Gal-3 is a promising biomarker that may identify potential patients of coronary atherosclerosis.
Fatty Liver Disease and Fibrosis	▪ Liver injury → Kuppfer cells and hepatocytes produce cytokines ▪ Cytokines activate stellate cells in disse space. ▪ Stellate cells transform into myofibroblast-like cells, which are capable of producing collagen pro-inflammatory cytokines and mediators. ▪ Hepatocellular damage tissue fibrosis.	Gal-3	▪ Most Gal-3 are firmly related to the pathologic procedures in fatty liver infection and fibrosis. At the point when focused restraint of gal-authoritative in people has been accomplished, it will be an extraordinary method for treatment. Numerous organizations are creating medications that are intended to target Gal-3 and restrain its coupling function(s).
Multiple Sclerosis (NDD)	▪ Blood–Brain barrier breaks down and, thus, allows the T-cells into the nervous system and decreases the integrity of tight junctions ▪ Immune system attacks the nervous system and destroys oligodendrocytes, causing demyelination ▪ T-cell attacks on myelin triggers inflammatory processes, thus, causing swelling and more activation of cytokines and other proteins	Gal-1 Gal-3 Gal-4 Gal-9	▪ Gal-1, Gal- 3, and Gal-9 are available at discernible levels in CWM, and, curiously, are fundamentally improved in dynamic MS injuries. On the cell level, galectins are confined to microglia or macrophages, astrocytes, and endothelial cells. ▪ In specific societies, brought by the arrival of endogenous Gal-4 connected with the beginning of myelination and the exogenous expansion of Gal-4, emphatically repressed myelination. This demonstrates neuronal Gal-4 emission and likely decides the planning of myelination by averting untimely myelination.
Amyotrophic Lateral Sclerosis (NDD)	▪ It affects the upper and lower motor neurons ▪ Destruction of these motor neurons disrupts the communication between the nerve and muscle ▪ Muscles atrophy and exhibit twitching	Gal-1 Gal-3	▪ Gal-1 was identified in axonal spheroids of motor neurons, a pathological hallmark of human ALS, but its mechanism on spinal motor neurons is not clear, though it does show potential therapeutic effect. ▪ Rise of Gal-3 level gives off the impression of being demonstrative of the beginning of ALS indications. Given its secretory nature and recognizable level in CSF, it is a potential biomarker for ALS that is deserving of further assessment as a treatment for ALS.

## Data Availability

No public data used, no data to report.

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
