# Peer review of "Galectins—Potential Therapeutic Targets for Neurodegenerative Disorders"

_ijms, 2022, doi:10.3390/ijms231911012_

Round 1
Reviewer 1 Report
In this review article "Galectins- Potential Therapeutic Targets for Neurodegenerative Disorders" the authors have comprehensively presented literature review of the Galectins to be used as potential targets for therapeutic intervention against neurodegenerative disorders. I think the paper has enough data to be published. Additionally, I found the figures and tables quality satisfactory.
Below are some genernal minor comments that should be addressed before publication.
1. The abstract is very short and proper rationale of the review, its significance and objective should be clearly presented.
2. Overall, English language of the article is good but need a through review to reduce some grammatical errors.
3. I have not seen limitations of the study please add them in the conclusion section.
4. The inclusion and exclusion criteria is missing in the literature search.
Author Response
We want to thank the reviewers for their positive comments and critique that helped us improve our review article significantly. We have modified the manuscript by addressing the concerns raised by the reviewers. A point-by-point response to the queries is provided below. We have also uploaded a revised manuscript version with the modifications done with track changes.
Reviewer-1
In this review article "Galectins- Potential Therapeutic Targets for Neurodegenerative Disorders" the authors have comprehensively presented literature review of the Galectins to be used as potential targets for therapeutic intervention against neurodegenerative disorders. I think the paper has enough data to be published. Additionally, I found the figures and tables quality satisfactory.
We want to thank the reviewer for taking the time and evaluating our manuscript. The suggestions have significantly improved our review.
Below are some general minor comments that should be addressed before publication.
- The abstract is very short and proper rationale of the review, its significance and objective should be clearly presented.
Thank you for this comment. We have modified the abstract as suggested.
- Overall, English language of the article is good but need a thorough review to reduce some grammatical errors.
We have reviewed and corrected the English language as suggested.
- I have not seen limitations of the study please add them in the conclusion section.
Thank for this comment. We have now included the limitation of use of Galectins as potential therapeutics target in the conclusion section.
- The inclusion and exclusion criteria is missing in the literature search.
This is now included in the introduction section.
Reviewer 2 Report
My suggestions:
1. Table 1 is difficult to read. Column "Individual functions" may be better to align to the left margin or between margins.
2. I would add a figure, on how galectins could impact neurodevelopment.
3. Are galectins impact somehow Alzheimer's disease?
4. Are there any plasma or CSF markers of galectin-related neurodegenerations? Are galectins potential markers for neurodegeneration?
5. Can galectin genetic variants be risk factors for any form of neurodegeneration?
Author Response
Reviewer- 2
We want to thank the reviewer for taking the time and evaluating our manuscript. The suggestions have significantly improved our review.
- Table 1 is difficult to read. Column "Individual functions" may be better to align to the left margin or between margins.
Thank you for pointing this out. We have now revised the tables with proper margins.
- I would add a figure, on how galectins could impact neurodevelopment.
Thank you for suggesting this. We have now included an additional figure (Figure 3) to show the role of galectins in neurodevelopment.
- Are galectins impact somehow Alzheimer's disease?
Yes, this information is now added the introduction section.
- Are there any plasma or CSF markers of galectin-related neurodegenerations? Are galectins potential markers for neurodegeneration?
Yes, Gal-3 specifically has been identified as a biomarker in serum, plasma, and/or cerebrospinal fluid (CSF) in Alzheimer’s disease (AD), Parkinson’s disease (PD), and Amyotrophic lateral sclerosis (ALS). We have now included this information in the revised manuscript. Yes, galectins are potential biomarkers for neurodegeneration and our review tries to cover the same information.
- Can galectin genetic variants be risk factors for any form of neurodegeneration?
Yes, LGALS3 genetic variation in the Gal-3 encoding gene has been associated with poorer cognition in patients with AD (Trompet et. al. 2012) and systemic sclerosis (Cunha et. al. 2021). We have now included this information in the revised manuscript.
Round 2
Reviewer 2 Report
Thank you, the manuscript is acceptable now